# A Plausible Link of TMPRSS2/ACE2/AR Signaling to Male Mortality during the COVID-19 Pandemic in the United States

**DOI:** 10.3390/pathogens10111378

**Published:** 2021-10-26

**Authors:** Lilly M. Wong, Guochun Jiang

**Affiliations:** 1UNC HIV Cure Center, Institute of Global Health and Infectious Diseases, University of North Carolina at Chapel Hill, 130 Mason Farm Rd., Chapel Hill, NC 27599-7030, USA; lilly_wong@med.unc.edu; 2Department of Biochemistry and Biophysics, The University of North Carolina at Chapel Hill, Chapel Hill, NC 27599-7042, USA

**Keywords:** COVID-19, SARS-Cov-2, TMPRSS2, ACE2, testosterone, androgen receptor, ESR1, lung, intestine, prostate, breast

## Abstract

The COVID-19 pandemic continues around the world, where the United States is among the worst in terms of both morbidity and fatality of the viral infection. We aim to investigate the plausible link of tissue SARS-CoV-2 viral entry gene expression, such as TMPRSS2 and ACE2, with infection and death by gender during the COVID-19 pandemic in the United States. We find a significantly higher incidence of COVID-19 death in men than in women, even though SARS-CoV-2 infection in women is higher than in men. We discover that the expression of TMPRSS2 and ACE2 in intestine, but not in lung, tends to be positively associated with the incidence of SARS-CoV-2 infection in men. In contrast, the high incidence of death in men is negatively correlated with TMPRSS2/ACE2 expression in intestine. Strikingly, the correlation of TMPRSS2/ACE2 expression with SARS-CoV-2 infection and death is the opposite in females, compared with that in males. Interestingly, male hormone signaling seems to be involved in mortality, as the low expression of testosterone receptor AR in the prostate contributes to death in men according to age. These observations point to a plausible contribution of male hormone metabolism in the regulation of TMPRSS2/ACE2 signaling to fatality by SARS-CoV-2 infection in men.

## 1. Introduction

Severe acute respiratory syndrome coronavirus 2 (SARS-CoV-2) is the coronavirus responsible for the COVID-19 pandemic, which was declared by the World Health Organization (WHO) in March 2020 [1]. Its emergence started in December 2019 as pneumonia of unknown cause in 41 patients in Wuhan [1,2]. As of 11 October 2021, the WHO has confirmed over 219 million COVID-19 cases, with roughly 4.55 million deaths worldwide since its inception [3,4]. Members of the *Coronaviridae* family, a group of positive-sense, single-stranded RNA viruses, generally cause mild upper-respiratory infections such as the common cold. However, the previous emergence of two pathogenic beta-coronaviruses responsible for the SARS-CoV outbreak in Asia in 2003 and the Middle East respiratory syndrome (MERS-CoV) outbreak in Saudi Arabia in 2012 have resulted in severe disease and mortality, providing direction for COVID-19 studies [5,6].

The coronavirus genome is comprised of genes encoding several structural proteins: surface spike glycoproteins for essential interactions between cellular receptors and membrane fusion for cellular infection, envelope proteins and membrane proteins for the formation of the CoV envelope and the production of viral particles and nucleocapsid proteins for several regulatory functions, such as viral replication and packaging [5,7]. While these viral components are the risk factors, clinical and epidemiological characteristics for COVID-19 are not completely understood. Many of the trends revolve around increased age, gender, and underlying health conditions/co-morbidities such as diabetes, obesity, and cardiovascular disease [8,9]. A retrospective case analysis of a small cohort of 99 COVID-19-confirmed cases from the Huanan seafood market in Wuhan, China, indicates a high risk of developing severe or fatal respiratory disease in older males with comorbidities [10]. Other retrospective case studies have highlighted gender disparities in deaths related to COVID-19 as an epidemiological characteristic of COVID-19 [11,12].

However, the factors related to the gender disparity in COVID-19 deaths is unclear. A cohort study of 305 individuals ranging in ages 4 to 60 years old showed that the low expression level of nasal SARS-CoV-2 receptor angiotensin-converting enzyme 2 (ACE2) was related to the lower prevalence of SARS-CoV-2 infection in children. In contrast, a recent cohort study reported that there was no association between ACE2 protein activity in the bronchoalveolar lavage fluid and age. Furthermore, in the clinic, the use of ACE inhibitors (ACEIs) and angiotensin receptor blockers (ARBs) is not positively associated with improvement of SARS-CoV2 infection and COVID-19 death. In fact, the use of ACEIs or ARBs caused a slight increase in confirmed case rates and deaths [13]. These studies indicate that our current understanding of ACE2 in the context of the COVID-19 pandemic is premature, due to the data available being from relatively small cohorts.

With the most recent available data from approximately 32 million COVID-19 positive cases in the United States, we aimed to investigate the link of SARS-CoV2 infection and COVID-19 death to the expression of hormone receptor AR, ACE2 and its activation enzyme transmembrane peptidase serine 2 (TMPRSS2) in multiple matched ages of human tissues in vivo. We discovered that there was a plausible link between tissue TMPRSS2/ACE2/AR signaling and fatality in men during the COVID-19 pandemic.

## 2. Materials and Methods

### 2.1. SARS-CoV-2 Infection Cases and Death in the United States

Cases of SARS-CoV-2 infection and death are publicly available and were retrieved from the GenderSci Lab COVID Project [14]. Characteristics such as gender and age were used in the analysis from published CDC data as of 21 June 2021. Data from all 50 states of the United States of America and its two territories, Puerto Rico and US Virgin Islands, were reported. The data of infection in males or females were calculated from the total number of COVID-19+ infections in the US, while male or female COVID-19-related deaths were calculated from the overall total of COVID-19 associated deaths. The database does not include data for patients who tested negative for the virus.

### 2.2. Analysis of Tissue Gene Expression in Lung, Intestine, Prostate, and Breast In Vivo

Gene expression of TMPRSS2/ACE2/AR by age and gender was retrieved and analyzed based on the tissue RNA-seq open access data from The Human Protein Atlas (HPA) Program launched in September 2019 (version 19), which shows the distribution of proteins and RNA expression across all major tissues/organs in the human body [15]. Total RNA was extracted from tissue samples using the RNeasy Mini Kit (Qiagen, Germantown, MD, USA) [15]. Gene expression (protein transcripts per million, pTPM) was analyzed based on gender, including lung ACE2 (*n* = 258 for male; *n* = 126 for female), lung TMPRSS2 (*n* = 286 for male; *n* = 141 for female), intestine ACE2 (*n* = 84 for male; *n* = 51 for female), intestine TMPRSS2 (*n* = 84 for male; *n* = 51 for female), prostate AR (*n* = 152 for male), prostate ESR1 (*n* = 154 for male) and breast AR (*n* = 291 for female). These data were used as the age of these donors ranged from 20 to 79, which matches the age data of infection or death caused by SARS-CoV-2 and so are suitable for our statistical data analysis.

### 2.3. Statistical Analysis of Data

Infection and death analysis was performed by two-way student test, where *p* < 0.05 was considered significant. Correlation with tissue gene expression with infection or death was achieved by Excel linear regression analysis, where *p* < 0.05 was considered significant and 0.05 < *p* < 0.09 was considered a trend of significance.

## 3. Results

### 3.1. COVID-19 Causes Disparities of Both Infection and Death among Men and Women

In more than 32 million COVID-19 cases, more females were infected than males in the overall COVID-19+ cases (Figure 1A). Despite that, a significantly higher number of COVID-19 deaths occurred among males (52.80 ± 8.71%) compared with females (44.90 ± 7.58%) (Figure 1B). This was more significant when the COVID-19 death of each gender was adjusted by its COVID-19 rate among all COVID-19+ cases (Figure 1C), where the male COVID-19 death rate is 1.27 ± 0.17-fold higher than in females (Figure 1D). When examining the age range among both genders (Appendix A), infection rates were increased in the age range 0–29, but then started to decrease from ages 29–80 years old (Figure 2A). Different from the infection rate, death rate in both genders kept increasing from 4–84 years old (Figure 2B). Interestingly, overall, there was no significant difference of % SARS-CoV-2 infection between males and females among age periods (Figure 2C). However, in all the age periods, COVID-19-caused death was higher in the male than in the female population (Figure 2D), particularly in the ages between 0–4 and between 14–74 years old. Therefore, elder populations tend to die after SARS-CoV-2 infection, regardless of gender.

### 3.2. The Expression of TMPRSS2 and ACE2 in Tissues

Even though infection in females was higher than that in males, mortality in males was significantly higher than in females (Figure 1 and Figure 2). To determine a plausible mechanism for the high death incidence in men, we next investigated the association of tissue gene expression of viral entry receptors, TMPRSS2 and ACE2, and hormone metabolism receptor genes with genders. Tissue gene expression was measured by RNA-seq [15] (Appendix A). ACE2 expression (<2 pTPM) is quite low compared to all the other genes examined. In particular, ACE2 expression in the lung is much lower than in the intestine among either males or females. In male lungs, there was a trend of positive correlation between TMPRSS2 and ACE2 expression (*p* = 0.0855; Figure 3A). A significant positive correlation was discovered between TMPRSS2 and ACE2 expression in the intestine (*p* = 0.00364; Figure 3B). However, this was not discovered in the female lung or intestine (Figure 3C,D), indicating that TMPRSS2/ACE2 signaling may be more related to male than female development. To explore whether this is possible, we analyzed the association of TMPRSS2/ACE2 expression with male hormone receptor expression levels in tissues. We found a significant positive correlation between male androgen receptor (AR) expression in prostate with TMPRSS2 expression in the intestines (*p* = 0.00201; Figure 3E). A similar correlation pattern was observed between AR in prostate and ACE2 expression in intestine (*p* = 0.0145; Figure 3F). A correlation of male hormone receptor AR with either TMPRSS2 or ACE2 indicates that male hormone signaling may be involved in the activation of TMPSS2/ACE2 signaling in men.

### 3.3. Causal Link of TMPRSS2/ACE2/AR Tissue Expression with COVID-19 Cases and Death

Unlike SARS-CoV-2 infection, which reached its plateau at the age of 20–29, the incidence of death kept rising from ages 4–84 years old in both infected men and women (Figure 2A,B). To see whether there is any potential link between tissue TMPRSS2/ACE2 expression, morbidity and/or fatality of SARS-CoV-2 infection, we set out to analyze the correlation of gene expression with SARS-CoV-2 infection and COVID-19 death. RNA-seq data were derived from the HPA Program (Appendix A), which is the only tissue gene expression data suitable for our statistical data analysis with matched age and gender. Since no RNA-seq data were available for tissue gene analysis for people aged from 0–19, only gene expression data from people aged from 20 to 79 years of age were examined in this study. In the male, lung TMPRSS2 or ACE2 expression was not associated with SARS-CoV-2 infection. Surprisingly, intestinal TMPRSS2 (*p* = 0.0421) and ACE2 (*p* = 0.0199) expression positively correlated with SARS-CoV-2 infection in men (Figure 4A,B), which was also related to age. A positive correlation was observed between the expression of prostate androgen receptor AR (*p* = 0.0518) with SARS-CoV-2 infection by age in men (Figure 4C). Interestingly, the expression of ESR1, the nuclear receptor of female estrogen existed in men, was not correlated with SARS-CoV-2 infection. In fact, in female lung or intestine, a negative association was discovered between ACE2 expression and SARS-CoV2 infection (Figure 5A,B). Instead, the expression of breast AR, the receptor of male testosterone in women, significantly correlated with SARS-CoV2 infection (*p* = 0.0422) (Figure 5C). These data suggest that TMPRSS2/ACE2 expression has opposite impacts on SARS-CoV-2 infection between men and women.

Surprisingly, in lung, the expression of TMPRSS2 or ACE2 was not correlated with male death (Figure 6A). In contrast, the low expression of intestinal ACE2 (*p* = 0.00264) or TMPRSS2 (*p* = 0.00295) was correlated with a high level of COVID-19 death in men, which is seemingly related to age (Figure 6B). Since male COVID-19 death was much higher than female death, which kept increasing by age (Figure 2B,D), we reasoned that this death incidence in men may be associated with impaired male hormone signaling. We found that reduced male hormone signaling was involved in a high rate of male COVID-19 death, since higher levels of male hormone receptor AR (*p* = 0.0233) were correlated with lower COVID-19 death by age in men (Figure 6C). This correlation was not observed with the expression of ESR1, the receptor of the female hormone (Figure 6C). An opposite link was observed, since either lung or intestine ACE2 expression was positively correlated with COVID-19 death in women (Figure 7A,B). Strikingly, the expression of breast AR, the receptor of the male hormone, negatively correlated with death (*p* = 0.00879; Figure 7C). Together, these data indicate that a low level of TMPRSS2/ACE2 signaling in the intestine may account for the deaths in men, which may be related to the waning level of male hormone signaling during aging. However, this is not the case in COVID-19 deaths among the female population.

## 4. Discussion

ACE2, a receptor expressed by epithelial cells in organs such as the lung, breast, gut, prostate and testes, is bound by SARS-CoV-2 spike proteins (similar to SARS-CoV) for host cell entry [16,17]. In fact, our analysis showed a high level of ACE2 expression in the intestine compared with the lung. Tissue gene expression analyses indicate that there is a plausible association between TMPRSS2 and ACE2 in men, which are thought to be essential for SARS-CoV-2 entry in tissue targets such as the lung and intestine. Interestingly, our data indicate that there are disparities between SARS-CoV-2 infection and death among genders, i.e., females tend to be infected by COVID-19 while males tend to die after SARS-CoV-2 infection. It has been shown that SARS-CoV-2 can cause intestinal disturbances, which may directly impact ACE2 signaling or indirectly modulate local inflammatory response in the gut, which contributes to COVID-19 death [18]. With ACE2 as a viral entry receptor expressed by many tissues, it is also suggestive of an implication of the male reproductive system, which suggests further research into its correlation with testosterone secretion [19,20]. In addition to ACE2, TMPRSS2 is an androgen receptor cofactor sensitive to estrogen/androgen stimulation and has been established as a primer for SARS-CoV-2 spike proteins for increased infectivity [17]. TMPRSS2 is involved in several prostate cancers and is highly expressed in the prostate, where it is highly regulated by androgens and/or androgen receptors [21,22,23]. Our analysis supports a model in which the infection of SARS-CoV-2 in men is associated with TMPRRS2/ACE2 viral entry signaling, while a high incidence of death is associated with low levels of TMPSSR2/ACE2 in men. In both scenarios, AR hormone signaling seems to be involved. These are almost completely opposite to females, either in SARS-CoV-2 infection or in COVID-19 death, indicating that sex-related factor(s) such as hormone metabolism may be linked to a high incidence of deaths in males. It has been proposed that the gender differences in COVID-19 cases may be related to cigarette smoking, where smoking is generally more common among men than in women, and smoking may upregulate TMPRSS2/ACE2 in the lung [24]. Nevertheless, the current epidemiologic investigations of COVID-19 have not sufficiently linked tobacco use to sex disparity of COVID-19 cases and/or death [23]. While our analysis points to a link of hormone metabolism to COVID-19 death, whether it is directly related to age in males warrants further investigation. This is due to the fact that our data were limited by a relatively small number of intestine and prostate tissues used for RNA-seq analysis. This will need a large cohort study, ideally using a larger number of tissues from the same COVID-19 patients. It is worth noting that sex hormones also regulate the innate and adaptive immune responses, which may play an additional role in the severity of COVID-19, related to the disparity of deaths between male and female populations [25].

Due to the unavailability of tissue RNA-seq data from individuals <20 years old, mechanisms of infection and death in young children and young adults could not be investigated in this study. It has been reported that ACE2 expression in the lung is reduced in aged mice or rats [26,27]. Nasal ACE2 expression increases with age, which may be associated with low COVID-19 incidence in young children [28]. However, a recent geospatial study showed that the administration of ACEIs or ARBs failed to improve the incidence of COVID-19 confirmed cases and deaths. Instead, ACEIs or ARBs caused a higher number of deaths [13]. Here, our data support the idea that an agonist of TMPRSS2/ACE2 may improve COVID-19 death incidence in men; however, it may not be proper to administer it in female patients because ACE2 expression is positively associated with female COVID-19 death. Since testosterone/AR signaling is reduced during aging [19,29] and its reduction is associated with COVID-19 death incidence in men, testosterone therapy may be useful for boosting TMPRSS2/ACE2 signaling, and thereby reducing COVID-19 death incidence.

## Figures and Tables

**Figure 1 pathogens-10-01378-f001:**
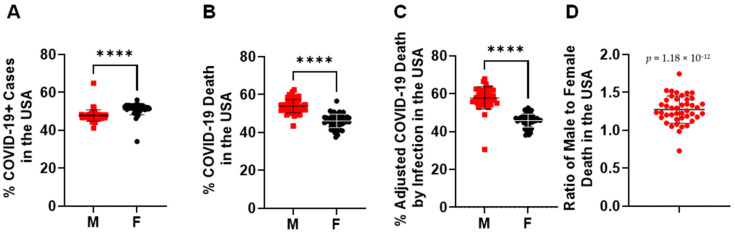
Gender disparities in SARS-CoV-2 infection and death in the United States. (**A**) SARS-CoV-2 infected more women than men in the total number of COVID-19+ cases. ****, *p* < 0.0001 (*n* = 50 states plus 2 territories). (**B**) SARS-CoV-2 infection caused more death in men than women in the total number of COVID-19 deaths. ****, *p* < 0.0001 (*n* = 50 states plus 2 territories). (**C**) % of COVID-19 death in each gender was adjusted by its infection rate among all COVID-19+ cases. ****, *p* < 0.0001 (*n* = 50 states plus 2 territories). (**D**) The death ratio of males to females was significantly higher than females alone in the total number of COVID-19 deaths, *p* = 1.19 × 10^−12^ (*n* = 50 states plus 2 territories). The morbidity or mortality rates were obtained by dividing the total number of COVID-19 + cases or COVID-19 deaths among women or men by the total number of individuals of that sex living in the USA.

**Figure 2 pathogens-10-01378-f002:**
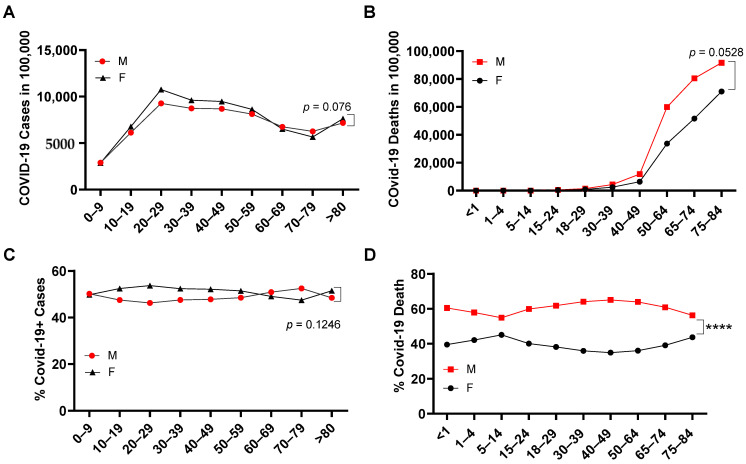
SARS-CoV-2 infection and death among ages in male and female. COVID-19+ cases (**A**) and COVID-19 death (**B**) per 100,000 people among ages in the United States (*n* = 50 states plus 2 territories). Gender percentage in overall COVID-19+ cases (**C**) and gender percentage in COVID-19 death (**D**) of each age range in the United States. ****, *p* < 0.0001 (*n* = 50 states plus 2 territories). The morbidity or mortality rates were obtained by dividing the total number of COVID-19+ cases or COVID-19 deaths among women or men by the total number of individuals of that sex living in the USA.

**Figure 3 pathogens-10-01378-f003:**
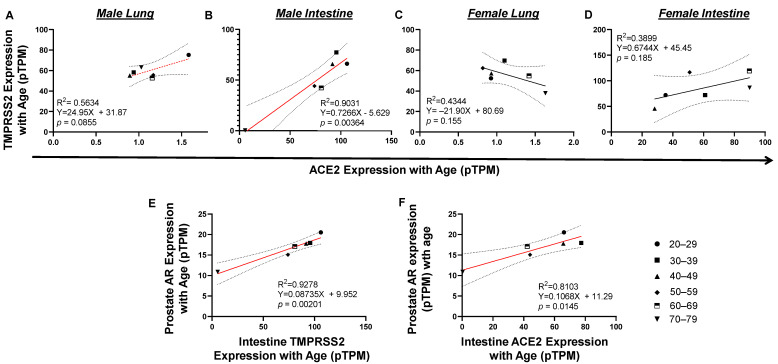
Tissue TMPRSS2/ACE2/AR expression correlated with each other by age in vivo. (**A** and **C**) There is a trend of TMPRSS2 expression correlated with ACE2 expression in male but not female lung. (**B** and **D**) Upregulation of TMPRSS2 significantly correlated ACE2 expression in male but not female intestine. (**E**) In prostate, upregulation of AR significantly correlated with TMPRSS2 expression. (**F**) In prostate, upregulation of AR significantly correlated with ACE2 expression. The linear regression was analyzed with the mean values of tissue gene expression, where *p* < 0.05 was considered significant and 0.05 < *p* < 0.09 was considered a trend of significance.

**Figure 4 pathogens-10-01378-f004:**
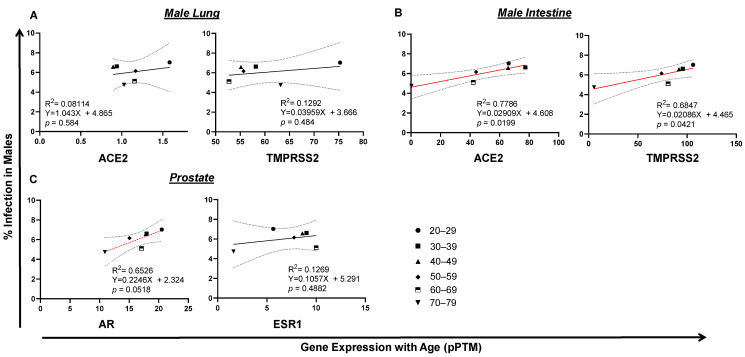
Tissue TMPRSS2/ACE2/AR gene expression correlates with SARS-CoV-2 infection in males. (**A**) No correlation of SARS-CoV2 infection with lung ACE2 expression in men. (**B**) Trends of intestinal TMPRSS2 or ACE2 expression with SARS-CoV-2 infection in men. (**C**) AR, but not ESR1, expression tended to correlate with SARS-CoV-2 infection in men. The linear regression was analyzed with the mean values of tissue gene expression vs SARS-CoV2 infection rate in the United States in males, where *p* < 0.05 was considered significant and 0.05 < *p* < 0.09 considered as a trend of significance.

**Figure 5 pathogens-10-01378-f005:**
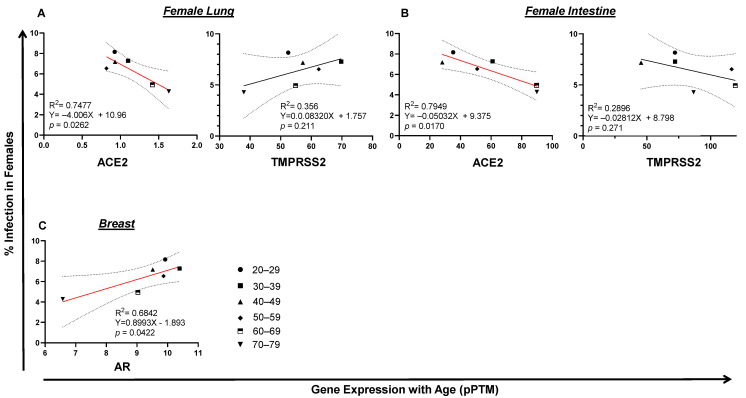
Tissue TMPRSS2/ACE2/AR gene expression correlates with SARS-CoV2 infection in females. (**A**) Lung ACE2 expression correlated with SARS-CoV2 infection in females. (**B**) Expression of intestinal ACE2, but not TMPRSS2, negatively correlated with SARS-CoV-2 infection in females. (**C**) Breast AR expression was associated with SARS-CoV-2 infection in females. The linear regression was analyzed with the mean values of tissue gene expression vs SARS-CoV2 infection rate in the United States in females, where *p* < 0.05 was considered significant.

**Figure 6 pathogens-10-01378-f006:**
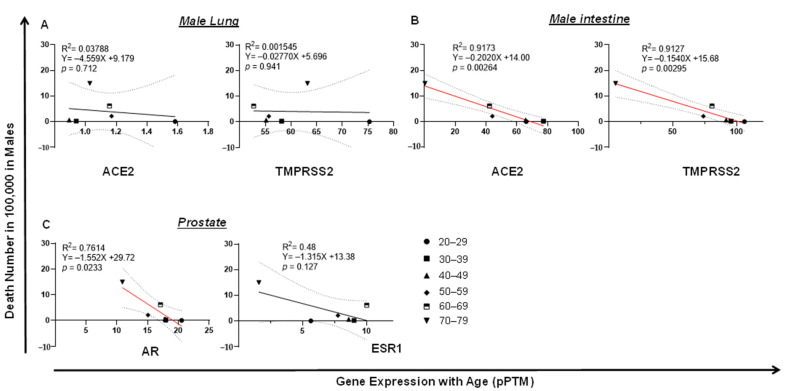
Tissue TMPRSS2/ACE2/AR expression negatively correlates with COVID-19 fatality in men. (**A**) No correlation of COVID-19 fatality with lung ACE2 or TMPRSS2 expression in men. (**B**) Downregulation of intestine TMPRSS2 or ACE2 expression significantly correlated with COVID-19 fatality in men. (**C**) Downregulation of prostate AR, but not ESR1, significantly correlated with COVID-19 fatality in men. The linear regression was analyzed with the mean values of tissue gene expression vs. COVID-19 male death rate in the United States, where *p* < 0.05 was considered significant.

**Figure 7 pathogens-10-01378-f007:**
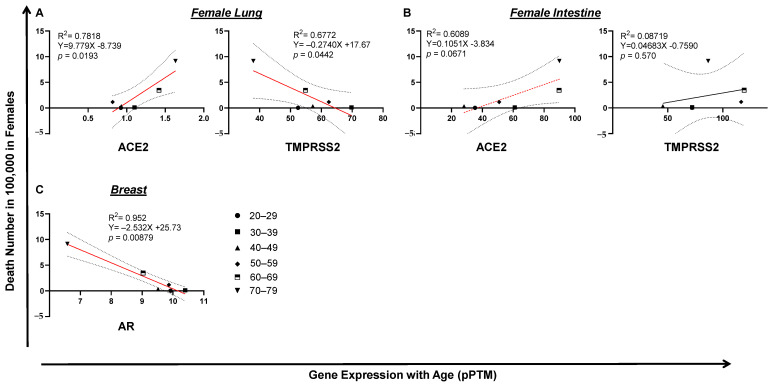
Tissue TMPRSS2/ACE2/AR expression and COVID-19 fatality in women. (**A**) Lung ACE2, but not TMPRSS2, expression correlated with COVID-19 fatality in women. (**B**) Upregulation of intestine TMPRSS2 or ACE2 expression significantly correlated with COVID-19 fatality in females. (**C**) Downregulation of breast AR significantly correlated with COVID-19 fatality in women. The linear regression was analyzed with the mean values of tissue gene expression *vs* COVID-19 female death rate in the United States, where *p* < 0.05 was considered significant and 0.05 < *p* < 0.09 considered as a trend of significance.

## Data Availability

The data have been supplemented in the manuscript.

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
