# Peer review of "A Plausible Link of TMPRSS2/ACE2/AR Signaling to Male Mortality during the COVID-19 Pandemic in the United States"

_pathogens, 2021, doi:10.3390/pathogens10111378_

Round 1
Reviewer 1 Report
In this brief report Wong and Jiang investigated the role of TMPRSS2/ACE2/AR in male mortality following infection with SARS-CoV-2. It is an exciting observation that although women are infected more often with SARS-CoV-2, more men die from COVID-19. So far, there is no real explanation for this observation. Therefore, it is quite exciting and useful for the community to take a closer look at this issue.
Here it is shown that there is a correlation between gene expression of ACE2 and TMPRSS2 especially in the intestine, but not in the lung, with death in men after SARS-CoV-2 infection. Interestingly, this correlation appears to be exactly reversed in women. Overall, the data analyses performed here show that male hormone metabolism in the regulation of TMPRSS2/ACE2 signaling is somehow related to male mortality after SARS-CoV-2 infection.
This observation is very interesting and sheds more light on the previously unanswered question of differential mortality rates between males and females after SARS-CoV-2 infection. The paper is well and concisely written and the analyses of the available data were plausibly performed. Overall, I think this report will add value to the SARS-CoV-2 community .
However, prior to publication, a few minor points should be addressed:
- The informations on SARS-CoV-2 in lines 31-32 should be updated, the informations presented here are still from the end of June 2021.
- Line 172: "...data from more than 32 million...": please remove the second "more than".
- The material and methods part is inserted twice in the manuscript, once after the discussion and once before the results. Please remove once.
Author Response
The informations on SARS-CoV-2 in lines 31-32 should be updated, the informations presented here are still from the end of June 2021.
Line 172: "...data from more than 32 million...": please remove the second "more than".
Response: We thank the Reviewer’s comments. We have updated the statement in the text to represent COVID-19 infection and death as of October 11, 2021.
The material and methods part is inserted twice in the manuscript, once after the discussion and once before the results. Please remove once.
Response: We thank the Reviewer’s comments and have removed the redundancy.
Reviewer 2 Report
Major Comments
- The authors do not explicitly state the disease status of the patients they analyzed that have been infected with SARS-CoV-2. Was this data collected from only patients that tested positive for viral infection? How specifically were patients selected to be included in this study or in the database referenced? Does the database also include data for patients who tested negative for the virus? The Material and Methods section should be greatly expanded upon in order to provide a detailed explanation of that database used.
- Figure 2: the statistical analysis used to determine the significance of correlations is not clear. What time points do the indicated p-values account for? The student’s t-test should have been performed at each time-point, and the corresponding p-values should be given for each time point, along with error bars for each point. As is, the figure does not intuitively indicate statistical significance of the authors’ findings.
- It is not clear why ACE2 expression in organs other than the lung are relevant to COVID-19-related mortality. The current model of SARS-CoV-2 infection implicates only the respiratory epithelium. Though there may be viral shedding in the digestive tract, which potentially implicates the intestine and prostate, ACE2 does not factor in at these sites because the virus does not cause infection outside of the respiratory epithelium. Additionally, ACE2 and TMPRSS2 expression have not been implicated in COVID-19 mortality, as they do not directly contribute to COVID-19 symptoms such as pneumonia and other systemic hyperinflammation. Taking into consideration the author’s hypothesis concerning hormone metabolism also does not explain their choice in organs, as the hormones in question are metabolized by processes that do not involve the prostate, intestine, or breasts. Unless the authors can clarify the importance of their selected organs (apart from the lungs), many of their findings seem irrelevant to COVID-19-related death.
- The authors conclude that low TMPRSS2/ACE2 expression in men predisposes them to COVID-19 mortality. It is unclear how such expression can logically be connected to mortality. With higher expression of these genes comes greater chance of viral infection in a larger number of host cells. This would potentially lead to increased immune activation and thereby severe COVID-19 symptoms. Outside of infectivity, TMPRSS2 and ACE2 have not been shown to alter the course of COVID-19 in patients. The authors should spend more time discussing their proposed model. Without proper justification, the model cannot be accepted or considered by the reader.
Minor Comments
Line 28: SARS-CoV-2 refers to the virus while COVID-19 refers to the diseases caused by infection by the virus. This point should be clarified.
Line 71, 76: The database from which data was retrieved should be cited or linked to in the main text.
In the results section, the way that values are reported from figure 1 are not clear. The numbers reported in parenthesis should be given the proper units (i.e. %) with the exception of fold change values.
Figure 1A: Were females actually infected at a higher rate? Or were there just more female patients with COVID than male patients in the database used? The authors should be precise in defining what it means when they say, “more female (49.19±10.16) were infected than male (46.00±9.57)” (Line 95).
Supplementary data seems to be missing from the manuscript PDF and therefore cannot be evaluated.
The authors present two identical materials and methods sections (Lines 268-290 and 68-90).
Previous studies have implied the importance of androgens and TMPRSS2 in COVID-19 and related disease, and should be addressed and/or referenced by the authors
https://cancerdiscovery.aacrjournals.org/content/10/6/779.abstract
https://link.springer.com/protocol/10.1007/978-1-0716-0211-9_2
https://www.mdpi.com/1422-0067/21/10/3627
Author Response
The authors do not explicitly state the disease status of the patients they analyzed that have been infected with SARS-CoV-2. Was this data collected from only patients that tested positive for viral infection? How specifically were patients selected to be included in this study or in the database referenced? Does the database also include data for patients who tested negative for the virus? The Material and Methods section should be greatly expanded upon in order to provide a detailed explanation of that database used.
Response: We apologize for the confusion. The data of infection in male or female were calculated from the total number of COVID-19 positive infections in the US while male or female COVID-19-related deaths were calculated from the overall total of COVID-19 associated deaths. The database does not include data for patients who tested negative for the virus. These were revised in the Figure legends and in the materials and methods.
Figure 2: the statistical analysis used to determine the significance of correlations is not clear. What time points do the indicated p-values account for? The student’s t-test should have been performed at each time-point, and the corresponding p-values should be given for each time point, along with error bars for each point. As is, the figure does not intuitively indicate statistical significance of the authors’ findings.
Response: We thank the Reviewer for bringing this to our attention. The data were cases of infection or % of death in male/female in each age range in the US. We only have overall infection and death data. We were not able to access the multiple datasets in each age range. Therefore, statistical analysis in the individual age range cannot be performed. Instead, the T-test was performed by comparing male or female across the age range.
It is not clear why ACE2 expression in organs other than the lung are relevant to COVID-19-related mortality. The current model of SARS-CoV-2 infection implicates only the respiratory epithelium. Though there may be viral shedding in the digestive tract, which potentially implicates the intestine and prostate, ACE2 does not factor in at these sites because the virus does not cause infection outside of the respiratory epithelium. Additionally, ACE2 and TMPRSS2 expression have not been implicated in COVID-19 mortality, as they do not directly contribute to COVID-19 symptoms such as pneumonia and other systemic hyperinflammation. Taking into consideration the author’s hypothesis concerning hormone metabolism also does not explain their choice in organs, as the hormones in question are metabolized by processes that do not involve the prostate, intestine, or breasts. Unless the authors can clarify the importance of their selected organs (apart from the lungs), many of their findings seem irrelevant to COVID-19-related death.
The authors conclude that low TMPRSS2/ACE2 expression in men predisposes them to COVID-19 mortality. It is unclear how such expression can logically be connected to mortality. With higher expression of these genes comes greater chance of viral infection in a larger number of host cells. This would potentially lead to increased immune activation and thereby severe COVID-19 symptoms. Outside of infectivity, TMPRSS2 and ACE2 have not been shown to alter the course of COVID-19 in patients. The authors should spend more time discussing their proposed model. Without proper justification, the model cannot be accepted or considered by the reader.
Response: As we described and discussed in the manuscript Introduction and Discussion, mounting evidence has shown that ACE2 expression is markedly higher in the small intestine under normal conditions (Livanos, A. E. et al. Intestinal host response to SARS-CoV-2 infection and COVID-19 outcomes in patients with gastrointestinal symptoms. Gastroenterology 2, 2 (2021)). In addition, a signifcant number of patients present with gastrointestinal symptoms, and high levels of viral RNA in the stool have been detected (Parasa, S. et al. Prevalence of gastrointestinal symptoms and fecal viral shedding in patients with coronavirus disease 2019: Asystematic review and meta-analysis. JAMA Netw. Open 3, e2011335 (2020).). This has been implicated with viral infection and death (Devaux, C. A., Lagier, J. C. & Raoult, D. New Insights Into the Physiopathology of COVID-19: SARS-CoV-2-Associated Gastrointestinal Illness. Front Med (Lausanne) 8, 640073, doi:10.3389/fmed.2021.640073 (2021); Pozzilli, P. & Lenzi, A. Commentary: Testosterone, a key hormone in the context of COVID-19 pandemic. Metabolism 108, 154252, doi:10.1016/j.metabol.2020.154252 (2020); Douglas, G. C. et al. The novel angiotensin-converting enzyme (ACE) homolog, ACE2, is selectively expressed by adult Leydig cells of the testis. Endocrinology 145, 4703-4711, doi:10.1210/en.2004-0443 (2004); Montopoli, M. et al. Androgen-deprivation therapies for prostate cancer and risk of infection by SARS-CoV-2: a population-based study (N = 4532). Ann Oncol 31, 1040-1045, doi:10.1016/j.annonc.2020.04.479 (2020).).
It was thought that male death may be associated with its hormone metabolism. In fact, a hormone-based clinical trial has been tested. However, the use of ACE inhibitors (ACEIs) and angiotensin receptor blockers (ARBs) is not positively associated with the rate of COVID-19 confirmed cases and deaths. In fact, the administration of ACEIs or ARBs caused a slight increase in confirmed case rates and deaths in patients (Montopoli, M. et al. Androgen-deprivation therapies for prostate cancer and risk of infection by SARS-CoV-2: a population-based study (N = 4532). Ann Oncol 31, 1040-1045, doi:10.1016/j.annonc.2020.04.479 (2020)). These studies indicate that, unlike previous thoughts, the induction of ACE2 expression may improve COVID-19 death. Our data are consistent with this recent new discovery which pointed to a negative association of ACE2/hormone signaling with Covid-19 male death. Male hormone levels decline during male aging. It is possible that ACE2 expression is associated with hormone metabolism, where low ACE2 expression is linked to high Covid-19 death in elder male. Our current study is the first step to understand this occurrence where future studies will be needed to test this phenomenon in cellular and animal models.
Minor Comments
Line 28: SARS-CoV-2 refers to the virus while COVID-19 refers to the diseases caused by infection by the virus. This point should be clarified.
Response: We thank the Reviewer’s comments and have clarified the terminology.
Line 71, 76: The database from which data was retrieved should be cited or linked to in the main text. In the results section, the way that values are reported from figure 1 are not clear. The numbers reported in parenthesis should be given the proper units (i.e. %) with the exception of fold change values.
Response: We thank the Reviewer’s comments. We have added the reference (Fagerberg, L. et al. Analysis of the Human Tissue-specific Expression by Genome-wide Integration of Transcriptomics and Antibody-based Proteomics. Mol Cell Proteomics 13, 397-406, doi:10.1074/mcp.M113.035600). Per the Reviewer’s comments, we have revised the writing by adding % in the value.
Figure 1A: Were females actually infected at a higher rate? Or were there just more female patients with COVID than male patients in the database used? The authors should be precise in defining what it means when they say, “more female (49.19±10.16) were infected than male (46.00±9.57)” (Line 95).
Response: The data were from overall Covid-19+ CONFIRMED cases. We have revised the sentence as “more female were infected than male in the overall Covid-19+ cases”.
Supplementary data seems to be missing from the manuscript PDF and therefore cannot be evaluated.
Response: The Supplementary data were placed before the References and after the Discussion section.
The authors present two identical materials and methods sections (Lines 268-290 and 68-90).
Response: We apologize for the error. We have removed the redundant materials.
Previous studies have implied the importance of androgens and TMPRSS2 in COVID-19 and related disease, and should be addressed and/or referenced by the authors
https://cancerdiscovery.aacrjournals.org/content/10/6/779.abstract
https://link.springer.com/protocol/10.1007/978-1-0716-0211-9_2
https://www.mdpi.com/1422-0067/21/10/3627
Response: We appreciate the Reviewer’s suggestions about these previous studies, which support the possible role of hormone metabolism in the regulation of ACE2/TMPRSS2 signaling we discovered in our study, leading the disparity of Covid-19 infection/death among genders. We therefore discussed these findings and cited their work, except the 2nd literature since it is a protocol only paper.
Reviewer 3 Report
In the other hand the research is poorly designed and the results are not significant. And the correlation between the results and the end point is incidental. The researchers try to answer an important issue in COVID-19 pandemic, but the model and methods did not really answer the questions. The results as such are neither clear nor sharp enough to reach conclusions about this point. There is need for improvement in the methods in general and to refine the results. The group has a wide database that can be used with proper planning of research questions and the right use of the right methods to produce a research platform that more accurately answers the proposed questions and provides more concrete answers.
Another comments:
Line 95. The division of patients into women and men should not be with a standard deviation.
Line 145-148 it does not clear who and how the samples were choose to make the RNA analysis
Line 149". Particularly, ACE2 expression in lung is much lower than in intestine either among male or female. In male lungs, there was a trend of positive correlation between TMPRSS2 with ACE2 expression (p=0.0855) (Figure 3A)". These results are not statistically significant.
Line 157: ". We found a significant positive correlation between male androgen re-ceptor (AR) expression in prostate with TMPRSS2 expression in the intestines (p=0.00201) (Figure 3E). A similar correlation pattern was observed between AR in prostate and ACE2 expression in intestine (p=0.0145) (Figure 3F). Taken together, these tissue gene expression analyses indicate that there is a plausible association between TMPRSS2 and ACE2, which are believed to be essential for SARS-CoV-2 entry in tissue targets such as lung and intestine"
Line 199" Taken together, these data indicate that a low level of TMPRSS2/ACE2 signaling in intestine may account for the death in men which may be related to the waning level of male hormone. However, this is not the case in the death among female population."
The results are indecisive and show a completely random relationship without evidence of a statistical relationship between the various results. On other hand, the author emphases that the invention of such a marker in men mediate infection but does not in women. Furthermore, the decrease in that marker is a predictor of mortality in men. These results are poorly understood from the article. The author complain that the entry of the COVID-19 virus is due the TMPRSS2 and ACE2. But the question of how this expression is emphasized in men causes the entry of the virus, while this lack of expression still explains the relatively high infection in women and high mortality rate in men.
Author Response
Line 95. The division of patients into women and men should not be with a standard deviation.
Response: We thank the Reviewer’s comments and have removed the standard deviation.
Line 145-148 it does not clear who and how the samples were choose to make the RNA analysis
Response: We thank the Reviewer’s critique. We researched the HPA database and have added the sentence “Total RNA was extracted from tissue samples using the RNeasy Mini Kit (Qiagen).”
Line 149". Particularly, ACE2 expression in lung is much lower than in intestine either among male or female. In male lungs, there was a trend of positive correlation between TMPRSS2 with ACE2 expression (p=0.0855) (Figure 3A)". These results are not statistically significant.
Response: We thank the Reviewer’s critical comments. As with the Reviewer’s opinion, we did not consider it as significant, instead, we stated that “there was a trend of positive correlation between TMPRSS2 with ACE2 expression.” This was also indicated in the materials and methods.
Line 157: ". We found a significant positive correlation between male androgen re-ceptor (AR) expression in prostate with TMPRSS2 expression in the intestines (p=0.00201) (Figure 3E). A similar correlation pattern was observed between AR in prostate and ACE2 expression in intestine (p=0.0145) (Figure 3F). Taken together, these tissue gene expression analyses indicate that there is a plausible association between TMPRSS2 and ACE2, which are believed to be essential for SARS-CoV-2 entry in tissue targets such as lung and intestine"
Line 199" Taken together, these data indicate that a low level of TMPRSS2/ACE2 signaling in intestine may account for the death in men which may be related to the waning level of male hormone. However, this is not the case in the death among female population."
The results are indecisive and show a completely random relationship without evidence of a statistical relationship between the various results. On other hand, the author emphases that the invention of such a marker in men mediate infection but does not in women. Furthermore, the decrease in that marker is a predictor of mortality in men. These results are poorly understood from the article. The author complain that the entry of the COVID-19 virus is due the TMPRSS2 and ACE2. But the question of how this expression is emphasized in men causes the entry of the virus, while this lack of expression still explains the relatively high infection in women and high mortality rate in men
Response: We thank the Reviewer’s insightful comments. Based on the currently available population data and scRNA-seq tissue database, this study discovered that COVID-19 caused more death in men than in women, where older males tend to a high degree of death compared with younger males. Our data pointed to a causal link of male Covid-19 death to tissue male hormone signaling. Further, we discover that the expression of TMPRSS2 and ACE2 in intestine, but not in lung, tends to be positively associated with the incidence of SARS-CoV-2 infection in men. In contrast, the high incidence of death in men is negatively correlated with TMPRSS2/ACE2 expression in intestine. Interestingly, the correlation of TMPRSS2/ACE2 expression with SARS-CoV-2 infection and death are opposite in female, compared with that in male. Further, the male hormone signaling seems involved in the mortality as the low expression of testosterone receptor AR in the prostate contributes to the death in men by age. We agree with the Reviewer that this is an observational study which is the first step for us to understand the high incidence of male COVID-19 death. This is due to the limited patient tissue samples currently, such as intestine and prostate. We hope that future studies will overcome this hurdle where much solid analysis will be achieved.
Round 2
Reviewer 2 Report
I believe that the revisions that the authors have made based on my comments are sufficient.
Author Response
We thank the Reviewer's comments. As suggested by the Editors for the language, we have revised updated the manuscript, which is highlighted in the new version.